# AMBIGUITY IN LLMS IS A CONCEPT MISSING PROBLEM

## ABSTRACT

Ambiguity in natural language is a significant obstacle for achieving accurate text to structured data mapping through large language models (LLMs), which affects the performance of tasks such as mapping text to agentic tool calling and text-to-SQL queries. Existing methods to ambiguity handling either rely on the ReACT framework to obtain correct mappings through trial and error, or on supervised fine-tuning to bias models toward specific tasks. In this paper, we adopt a different approach that characterizes representation differences of ambiguous text in the latent space and leverages these differences to identify ambiguity before mapping them to structured data. To detect sentence-level ambiguity, we focus on the relationship between ambiguous questions and their interpretations. Unlike distances calculated by dense embeddings, we introduce a new distance measure based on a path kernel over concepts. With this measurement, we identify patterns to distinguish ambiguous from unambiguous questions. Furthermore, we propose a method for improving LLM performance on ambiguous agentic tool calling through missing concept prediction. Both achieve state-of-the-art results.

## 1 INTRODUCTION

Question answering using large language models (LLMs) often fails when user questions are ambiguous. A growing strand of work like (Min et al., 2020; Stelmakh et al., 2022) shows that a surprising fraction of their "errors" can be traced back, not due to lack of knowledge in LLM, but to ambiguity in the user's question itself. This ambiguity does not just mean that the question does not provide enough information, but also that the question has ambiguous semantics, i.e., multiple interpretations.

Existing studies focus more on pragmatic or lexical ambiguity, ambiguity handling in these studies either exploits the ReACT(Yao et al., 2023) framework to produce correct mappings through trial and error, or supervised fine tuning to guide models to produce biased mappings to improve on certain tasks(Saparina & Lapata, 2025). Kamath et al. (2024) attempt to use LLMs to detect ambiguity of sentences whose meaning changes with the relative scope of quantifiers, negation, or modals. They show that powerful LLMs trained on the most comprehensive datasets, such as GPT-4 sometimes default to a non-preferred semantic reading, and that success of disambiguating text varies sharply with different phrasing, which indicates disambiguation can not be easily solved by LLMs themselves. The ambiguity detection results in (Saparina & Lapata, 2024) also confirm this observation. On the other hand, there is limited research on representational differences of ambiguous text. In this work, we study the representation of ambiguous text in the latent space and leverage the differences to identify ambiguity.

As an ambiguous utterance has multiple interpretations, studying the distribution of interpretations is a natural way for ambiguity detection (Stengel-Eskin et al., 2023). Figure 1 provides an example of the relationships between the ambiguous query $q$ and its corresponding two interpretations, denoted by $i_1$ and $i_2$. Ideally, a good distance measurement may uncover the pattern of the triplet associated with an ambiguous utterance. Unfortunately, current distance measurement by dense embedding vectors(Karpukhin et al., 2020) cannot give us such a measurement. The distances computed by dense vectors focus more on the semantics of individual words than on the structure of the entire sentence, which is not sensitive to the ambiguity caused by the structure of the sentence, particularly when some concepts are missing in the sentence.

In our study, we observe that the ambiguity is often associated with missing concepts in the input utterances. With the recent progress on LLM interpretability (Bricken et al., 2023; Templeton et al.,

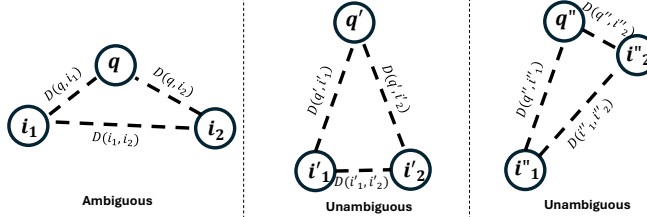

Figure 1: An example to show the difference of the distance measurement on triplets for ambiguous question $(q, i_1, i_2)$ and two kind of unambiguous question $(q', i_1', i_2')$, $(q'', i_1'', i_2'')$. For similar queries ambiguous $q$ and unambiguous $q'$, $q''$, we expect $\overline{(\mathrm{D}(q, i_1), \mathrm{D}(q, i_2), \mathrm{D}(i_1, i_2))} > \overline{(\mathrm{D}(q', i_1'), \mathrm{D}(q', i_2'), \mathrm{D}(i_1', i_2'))}$ and $|\mathrm{D}(q, i_1) - \mathrm{D}(q, i_2)| \ll |\mathrm{D}(q'', i_1'') - \mathrm{D}(q'', i_2'')|$ (the overline means average).

2024), the human-understandable concepts embodied in a utterance can be extracted together with their representation in the latent space through a sparse autoencoder (SAE). This inspires us to design methods to learn the concept differences of multiple interpretations of the input utterance in the latent space to identify ambiguity. We further leverage a kernel method (Domingos, 2020) to develop a distance metric for such latent concept comparison. We turn SAE into a kernel machine to measure the similarity between the concept representations of different interpretations of an ambiguous utterance. The computing is done through the integral of gradient values in a path kernel for each concept extracted by SAE. To make the similarity measurement focus on the target semantic patterns, we filter out concepts irrelevant to input utterances. By doing this, we successfully discover the pattern of ambiguous questions.

Once ambiguous utterances are identified, incorporating additional information can improve their mapping to structured data. When the structured data requires intermediate such as SQL generation to access, users often need to be asked to clarify the utterances and provide additional information. When the structured data are finite and well-defined, e.g., tools defined within an agentic framework, this additional information can be obtained from the concepts embodied in the data. We exploit the difference of concepts between ambiguous queries and target structured data, and design a missing concept prediction model to assist the mapping. We show in the experiments that our method achieves the best API calling performance on Gorilla(Patil et al., 2024) TensorFlow Hub bench.

In summary, our work make the following contributions:

1. We observed that ambiguity arises from *missing concepts* in the latent space of LLMs (Section 3.1 and 3.2). Using this insight, we designed a new distance measure that enhances interpretability and targets specific semantic patterns.

2. We identify patterns to distinguish ambiguous from unambiguous questions with this measurement.

3. We propose a new framework to enhance the performance of LLMs in handling ambiguous agentic tool calls by predicting missing concepts.

## 2 PRELIMINARY

**Path Kernel.** Path kernels are used to measure how similarly a model at two data points varies during learning. Here we refer to the explanation for kernel machine from (Domingos, 2020), a *kernel machine* predicts

$$y = g\Big(\sum_i a_i\, K(x, x_i) + b\Big),$$

with the kernel $K$ measuring the similarity between data points. Gradient-descent training (learning rate $\varepsilon \to 0$) implies that the final predictor behaves like a **path kernel** machine:

$$\mathrm{K}_{\text{path}}(x, x') = \int_{c(t)} \nabla_w y(x) \cdot \nabla_w y(x')\, dt,$$

where $c(t)$ is the parameter trajectory during training. The more aligned the gradients of $y$ at $x$ and $x'$, the larger the kernel value, thus the variations of $x$ and $x'$ are more similar during training.

**Sparse autoencoder (SAE).** Neurons in modern language models often behave such that the same unit fires for several unrelated concepts. A leading hypothesis (Superposition Hypothesis) is (Elhage et al., 2022): the model stores many more features than it has neurons by packing them into an over-complete set of directions in activation space. Recovering those directions is therefore a natural route to mechanistic interpretability. The work by Anthropic shows that a *sparse auto-encoder* (SAE) trained directly on a layer's activations can do exactly this, yielding thousands of highly interpretable, near-monosemantic concepts(Bricken et al., 2023).

Let $\mathbf{H}(x) \in \mathbb{R}^d$ denote the hidden-state (e.g. residual-stream) vector produced by an LLM for a token sequence $x$. The goal is to learn a *dictionary* $\{\mathbf{d}_i\}_{i=1}^n \subset \mathbb{R}^d$ such that every activation can be reconstructed from a **sparse** combination of these directions:

$$\mathbf{H}(x) \;\approx\; \mathbf{b} \;+\; \sum_{i=1}^{n} f_i\big(\mathbf{H}(x)\big)\,\mathbf{d}_i \tag{1}$$

in which, $\mathbf{b} \in \mathbb{R}^d$ is a learned bias that captures the mean activation. $f_i(\mathbf{H})$ is a *gate* that decides whether feature $i$ is present; the ReLU promotes non-negativity and sparsity: $f_i(\mathbf{H}) = \mathrm{ReLU}\big(\langle \mathbf{w}_i, \mathbf{H}\rangle + b_i^{\mathrm{enc}}\big)$; $\mathbf{d}_i$ is the **decoder** vector that take the feature back into the original space.

After training, each dictionary row $\mathbf{d}_i$ corresponds to a *concept*. These recovered concepts are **Sparse** (only a handful activate per token), **Linear** (they live directly in the model's latent space), and **Monosemantic** (each gate corresponds to one dominant pattern); as a result, they can be inspected, clamped, and ablated far more easily than raw neurons.

**Ambiguity in NLP.** Ambiguity has been studied across various NLP tasks including machine translation (Pilault et al., 2023), natural language inference (Liu et al., 2023), question answering (Kim et al., 2024; Sun et al., 2023), and semantic parsing (Mu et al., 2024; Saparina & Lapata, 2025). Recent approaches leverage LLMs to detect ambiguities by sampling multiple candidate solutions and resolving ambiguities through clarification questions or by prompting alternative interpretations. For instance, Mu et al. (2024) samples multiple outputs from an LLM and examines their consistency to identify potential ambiguities. When inconsistencies are detected, the LLM is prompted to generate targeted clarification questions. However, due to the inherent biases of LLMs, the sampled solutions may lack diversity, making some ambiguities difficult to detect. To address this limitation, Saparina & Lapata (2025) generates an initial set of default interpretations using an LLM, which are then augmented using a specialized infilling model that requires supervised training. Our work instead examines ambiguity in the latent concept space.

## 3 METHODOLOGY

In this section, we first define the ambiguity resolution problem as a missing concept problem (3.1). We then show the effect of adding missing concepts(3.2), followed by describing our ambiguity detection method (3.3). Finally we describe how we predict missing concepts in the context of agentic tool calling (3.4).

### 3.1 AMBIGUITY RESOLUTION AS A MISSING CONCEPT PROBLEM

LLMs often bias towards generating one interpretation among many for an ambiguous utterance (Saparina & Lapata, 2025). Prompting LLMs to produce multiple interpretations and directly comparing their semantics do not help ambiguity detection. To show LLMs do not produce different interpretations for an ambiguous utterance, we extract a sentence from the AMBROSIA dataset (Saparina & Lapata, 2024) and prompt Llama-3.3-70B-Instruct(Meta AI, 2024) to generate two interpretations for this sentence.

These two interpretations in fact have the same meaning. To trigger the generation of diverse interpretations, we exploit special tokens' role in steering LLMs' responses. When we insert

*Ambiguous Question* (q): Show all Spanish-speaking gate agents and pilots.

Llama

*Interpretation 1*: Show all gate agents who speak Spanish and all pilots who speak Spanish.

*Interpretation 2*: Show all gate agents and pilots who speak Spanish.

"[MASK]" in the original sentence, the Llama model produces two interpretations aligned with the ground-truth:

*Perturbed Question (q′)*: Show all Spanish-speaking gate agents and [MASK] pilots.

Llama

*Interpretation 1*: Show all Spanish-speaking gate agents and all pilots.

*Interpretation 2*: Show all Spanish-speaking gate agents and all Spanish-speaking pilots.

Note that while "[MASK]" has no semantic meaning by itself, its presence in this position increases the sentence's uncertainty. To understand what changes are triggered by the "[MASK]" token in the concept space that make the model produce different interpretations. We use a sparse-autoencoder (SAE)(Goodfire, 2025) trained on the outputs of 50 layer of this Llama model to track the new concepts after the "[MASK]" token is inserted. We get the following key concept from the SAE:

*"Start of new paragraph or point in explanatory text".*

To verify the new interpretation is indeed triggered by this concept, we individually clamp the activation value of this concept (increase it to 10) while keeping the original input sentence unchanged (without inserting the "[MASK]" token). We obtain the following interpretations:

*Ambiguous Question (q):* Show all Spanish-speaking gate agents and pilots.

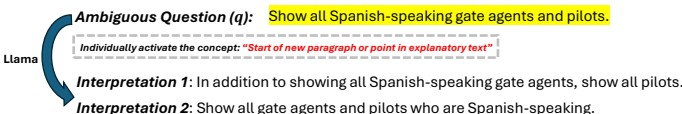

Llama

*Interpretation 1*: In addition to showing all Spanish-speaking gate agents, show all pilots.

*Interpretation 2*: Show all gate agents and pilots who are Spanish-speaking.

These interpretations match the ground-truth, which indicates an LLM can be steered to generate diverse interpretations for ambiguous utterances. We further show in the experiments that injecting examples in the prompt can effectively "remind" the model the missing concepts, therefore trigger the generation of diverse interpretations (see Appendix B.1 for examples). With the diverse interpretations of ambiguous inputs, we can then detect such ambiguity.

However, a naive approach of using the distances of dense vectors of generated interpretations to detect ambiguity does not work well. We use the following example to elaborate this. We denote the top two interpretations of the input utterance $q$ as $i_1$ and $i_2$. By using the output of the last hidden layer of Llama-3.3-70B-Instruct for the three sentences separately, we obtain the dense vectors of the triplets $(v(q), v(i_1), v(i_2))$:

*Perturbed Question (q′)*: Show all Spanish-speaking gate agents and [MASK] pilots.

*Interpretation 1 ($i_1$)*: In addition to showing all Spanish-speaking gate agents, show all pilots.

*Interpretation 2 ($i_2$)*: Show all gate agents and pilots who are Spanish-speaking.

Llama: $D(q′, i_1) > D(i_1, i_2) > D(q′, i_2)$

Although the "[MASK]" token activates additional concepts of the Llama model and makes it generate diverse interpretations, these additional concepts do not lead to sufficient changes in the dense vectors for ambiguity detection. The distance between $v(q′)$ and $v(i_1)$ is 0.17 and the distance between $v(q′)$ and $v(i_2)$ is 0.092, while the distance between $v(i_1)$ and $v(i_2)$ is 0.13. It is difficult to leverage the distance contrast in the triplet to derive a threshold to classify $q$ as ambiguous as the distance between interpretations can be arbitrarily smaller. We have done experiments with advanced embedding models and they do not have satisfactory sensitivity for distinguishing ambiguity patterns either.

However, we notice that $q′$ and $i_1$ activated some concepts in common, which inspires us to utilize the concept differences of the triplet in the latent space to detect pattern of ambiguity. We show that such distance measure can produce sufficient sensitivity for ambiguity detection. The distances measured using our method are as follows: $D(q′, i_1) = 0.039$; $D(q′, i_2) = 0.027$; $D(i_1, i_2) = 0.043$, meaning

the distance between interpretations is larger than their distances to the query. This property produces a sensitive metric for ambiguity detection. We explain the proposed distance metric in Section 3.3.

## 3.2 Effect of Adding Missing Concepts into the Latent Space

To further explore the relationship between semantic ambiguity (uncertainty in LLMs) and missing concepts, we use semantic entropy(Kuhn et al., 2023) to measure the ambiguity of query semantics (see Appendix A for algorithm details). We compute the semantic entropy produced by LLama-3.3-70B-Instruct (Meta AI, 2024) on 1) 20 ambiguous questions; 2) the same 20 questions with random concepts activated; and 3) the same 20 questions with missing concepts activated.Figure 2 shows that without any additional concept activated, the semantic entropy is close to zero, indicating the LLM produces a single interpretation on the input. This explains the semantic entropy of queries produced by Llama alone cannot detect

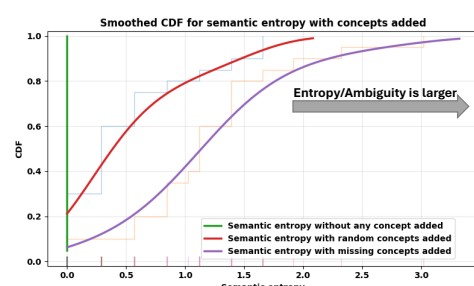

Figure 2: Entropy/ambiguity change with missing concepts added.

ambiguity. With the missing concepts activated, the interpretations become diverse, evidenced by the increase of semantic entropy of queries. Activating random concepts also increases the semantic entropy because the noise introduced leads to diverse semantics, but the increase is less significant than that caused by adding target missing concepts.

This explains the effect of background knowledge on semantic ambiguity. For example, for this question: "Who won the war between ethiopia and italy?", if the LLM lacks the context of Italo-Ethiopian War, it does not know where the ambiguity is. Once the LLM retrieves the context of Italo-Ethiopian War from external sources, the concept space is enriched with the "First War" or the "Second War", which in turn increases the semantic entropy of the question.

This also explains why fine-tuning works on ambiguity resolution. Fine-tuning can be seen as learning to activate the missing concepts and therefore increase the semantic entropy of ambiguous queries.

## 3.3 Representation-based Ambiguity Detection

Our solution (see Figure 3) is to use the path kernel with a sparse autoencoder (SAE) as the kernel machine for calculating distances between data points.

**SAE as Kernel Machine.** We consider SAE as $y$, the input sentences are $x$ and $x'$, their hidden states on LLM's layer where the SAE trained on are $H(x)$ and $H(x')$. Therefore, we have

$$\mathrm{K}(x, x') \;=\; \int_{c(t)} \big(\nabla_{\mathbf{w}}\mathrm{SAE}(H(x))\big) \cdot \big(\nabla_{\mathbf{w}}\mathrm{SAE}(H(x'))\big)\, dt \tag{2}$$

Here, we denote the SAE on the given concept dictionary as $\mathbf{f}_{\mathrm{SAE}}$, where $\mathbf{f}_{\mathrm{SAE}}(\mathbf{H}(\mathbf{x})) = (f_1(\mathbf{H}(\mathbf{x})), \ldots, f_N(\mathbf{H}(\mathbf{x})))^\top$, ($f(\mathbf{H})$ in Equation 1 ). The $i$-th activation is then simply $f_i(\mathbf{x})$.

$$\mathbf{f}_{\mathrm{SAE}}(\mathbf{H}(\mathbf{x})) \;=\; \mathrm{ReLU}\big(W_e(\mathbf{H}(\mathbf{x}) - \mathbf{b}_d) + \mathbf{b}_e\big) \;\in\; \mathbb{R}^N, \tag{3}$$

where $W_e$ is the weight matrix of the encoder and $\mathbf{b}_d, \mathbf{b}_e$ are a pre-encoder and an encoder bias, respectively.

Not all $N$ concepts are necessary for the path kernel calculation. To obtain the variation for input sentences $x$ and $x'$, we only need to focus on the target concepts. Accordingly, we apply a mask $M$ on the features used in gradient computation when calculating the path kernel:

$$\mathrm{K}(x, x') = \int_{c(t)} \nabla_{\mathbf{w}}\mathbf{f}_{\mathrm{SAE}}^{\mathrm{mask}}\big(\mathbf{H}(x)\big) \cdot \nabla_{\mathbf{w}}\mathbf{f}_{\mathrm{SAE}}^{\mathrm{mask}}\big(\mathbf{H}(x')\big)\, dt, \tag{4}$$

$$\mathbf{f}_{\mathrm{SAE}}^{\mathrm{mask}}(\mathbf{H}(x)) = M \circ \Big[\mathrm{ReLU}\big(W_e\,(\mathbf{H}(x) - \mathbf{b}_d) + \mathbf{b}_e\big)\Big] \tag{5}$$

where $M$ is the concept mask (explained below) and $\circ$ is hadamard product.

**Determining Unmasked/Target Concepts.** Semantic distances between two sentences are typically measured using the cosine similarity of their dense vectors generated by large embedding models (Muennighoff et al., 2022). However, this distance measurement is not sensitive to the semantic distinctions we want.

In interpretation generation, we use concept embodied examples (see Appendix B.1) for triggering the generation of diverse interpretations. To ensure our distance calculation captures the semantic meaning of sentences, we distill the concepts activated by their semantics and restrict the path kernel computation to these concepts. The distillation process involves three steps:

1. Collect the concepts activated by the example triplet sentences by LLM with SAE.
2. Remove the concepts activated by each individual token $t_i$ in the example triplet sentences from the set of concepts recorded in step 1.
3. Include the remaining concepts in the mask vector $M$, which are considered valid:

$$M = \{\mathbf{f}(\mathbf{H}(\mathbf{x}))\} \setminus \{\mathbf{f}(\mathbf{H}(\mathbf{t_1})), \dots, \mathbf{f}(\mathbf{H}(\mathbf{t_n}))\} \quad (6)$$

Here $x$ is the example sentence and $t_1, \dots, t_n$ are the tokens in the sentence.

**Path State Approximation.** We use a path kernel to characterize relationship of the obtained latent representations of concepts. Path states are the snapshots of a model's parameters saved after each optimization step during training or fine-tuning. For a pre-trained SAE we usually only have the final weights, so the original series of path states cannot be reconstructed exactly. When re-training is impossible or costly, we can replace the unknown gradient-descent path with a straight-line interpolation in parameter space.

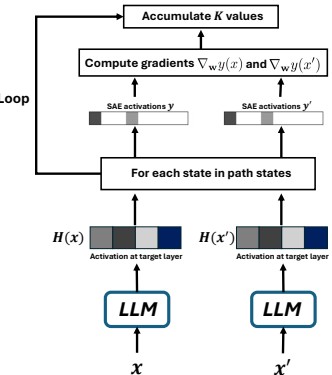

Figure 3: The workflow of the path kernel calculation with SAE.

Let

$$\Theta = \{\theta_k\}_{k=1}^P, \Theta^{(0)} = \{\theta_k^{(0)}\}_{k=1}^P, \Theta^* = \{\theta_k^*\}_{k=1}^P$$

be, respectively, the parameter set, the (zero) initialization, and the final pre-trained weights.

By choosing $n$ interpolation steps and define $\alpha_j = \frac{j}{n-1}, \quad j = 0, 1, \dots, n-1.$, the $j$-th intermediate snapshot is then

$$\Theta^{(j)} = (1 - \alpha_j)\,\Theta^{(0)} + \alpha_j\,\Theta^*, \quad \theta_k^{(j)} = (1 - \alpha_j)\,\theta_k^{(0)} + \alpha_j\,\theta_k^*$$

Collecting them gives the full set of path states:

$$\{\Theta^{(0)}, \Theta^{(1)}, \dots, \Theta^{(n-1)}\}.$$

Here, $\alpha$ increases linearly from 0 to 1, forming a straight-line path. Although this sequence does not follow the true gradient-descent dynamics, it provides a simple, deterministic path that is often adequate for estimating a path kernel.

**Distance Measurement.** The path kernel measures how similar two data points are according to the model based on their changing trajectories along the paths. To convert the (unnormalized) path kernel $K(\cdot, \cdot)$ into a proper distance metric between data points $x$ and $x'$, we apply the following two standard normalizations:

$$D_1(x, x') = 1 - \frac{K(x, x')}{\sqrt{K(x, x)\,K(x', x')}}, \quad (7)$$

$$D_2(x, x') = \sqrt{K(x, x) + K(x', x') - 2\,K(x, x')}, \quad (8)$$

We show in the experiments that both $D_1$ and $D_2$ can identify ambiguity and can be used for serving different objectives.

## 3.4 PREDICTING MISSING CONCEPTS TO MITIGATE AMBIGUITY

In Section 3.1, we argue that the ambiguity problem arises from concepts missing in LLM's latent space, and thus distance measurements should be sensitive to this. Based on this hypothesis, we propose using path kernels with SAE to measure distances between questions and their interpretations. As shown in our experiments, this method reveals patterns that distinguish ambiguous questions from unambiguous ones. Motivated by this, we investigate whether ambiguity can be exploited to reduce incorrect responses and better align outputs with training data. To this end, we introduce a framework - within the context of tool calling - that retrieves data chunks by training a concept predictor on labeled data.

As illustrated in Figure 4, instead of using dense embedding vectors to retrieve API calls (as in (Karpukhin et al., 2020)), we first collect the concepts activated by questions and documents on LLM by its SAE. We then use the trained concept predictor to predict missing concepts in input questions. Finally, we rank API calls using union joint based on concept matching. Appendix C.1 presents examples of ambiguous questions in tool calling, while Appendix C.2 illustrates concept matching in this context.

For efficiency, we use LightGBM (Ke et al., 2017) to train the concept predictor. For each concept activated by the training data and documents, the predictor is trained to determine whether it is missing from the input question:

$$p(y = 1 \mid x) = \sigma\Big(\sum_{t=1}^{T} \eta\, f_t(x)\Big) = \frac{1}{1 + \exp\big(-\sum_{t=1}^{T} \eta\, f_t(x)\big)} \tag{9}$$

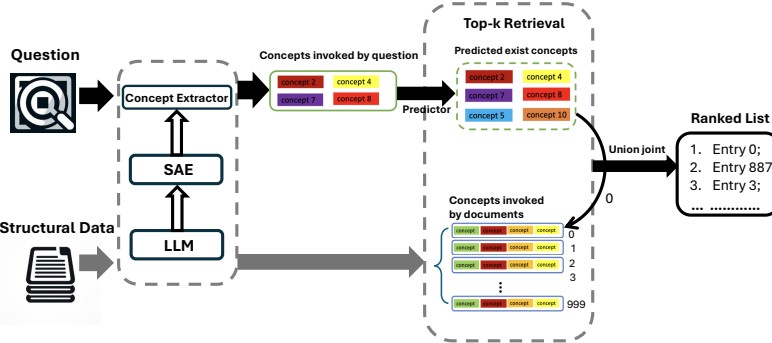

Figure 4: Tool calling framework based on missing concept prediction in ambiguous questions.

## 4 EXPERIMENTS

In this section, we conduct three sets of experiments:

1. Ambiguity detection by the proposed distance metrics: We investigate whether the distances between questions and their interpretations can distinguish ambiguous questions from unambiguous ones.
2. Ambiguity sensitivity improvement of LLMs: We show if adding missing concepts can improve LLM's self-judgment on ambiguous questions. (see Appendix B.3)
3. Ambiguity resolution on agentic tool calling: We investigate whether predicting missing concepts in ambiguous questions can reduce the number of incorrect responses.

## 4.1 AMBIGUITY DETECTION

**Experiment Settings.** To evaluate the effectiveness of our method for ambiguity detection by distance differences, we conduct experiments primarily on AMBROSIA (Saparina & Lapata, 2024) [1], a benchmark designed for parsing ambiguous questions into database queries across multiple domains. The

---

[1]We also use another ambiguous dataset ASQA (Stelmakh et al., 2022), reported in Appendix B.3.

benchmark consists of 1,277 ambiguous questions, each paired with human-provided unambiguous interpretations and corresponding SQL queries (2,965 in total), spanning 846 multi-table databases across 16 distinct domains. It includes three types of ambiguity—scope ambiguity, attachment ambiguity, and vagueness—and goes beyond earlier datasets that assume a single "correct" query, offering a rigorous evaluation yardstick for models that must both detect ambiguity and enumerate all valid SQL programs.

We first prompt LLMs to generate interpretations for the ambiguous questions in the dataset. Specifically, we use LLama-3.3-70B-Instruct (Meta AI, 2024) for this task (see Appendix B.1 for a prompting example). For each ambiguous question, we generate two interpretations, $i_1$ and $i_2$. These interpretations are then treated as unambiguous questions and are each further prompted to generate their own interpretations. As a result, for both ambiguous and unambiguous questions, we obtain two interpretations each, forming a triplet $(q, i_1, i_2)$.

Next, we compute the distances between the original question and its interpretations on AMBROSIA: $\mathrm{D}(q, i_1)$, $\mathrm{D}(q, i_2)$, and $\mathrm{D}(i_1, i_2)$. These distances are calculated using both our path kernel-based method and traditional dense vector-based methods. For comparing with both Embedding model and Generation models, we use SFR-Embedding-Mistral (Meng et al., 2024) and LLama-3.3-70B-Instruct to generate dense embeddings, with distances computed as follows:

$$\mathrm{D}(\mathbf{x}, \mathbf{x}') \;=\; 1 - \frac{\mathbf{E}(\mathbf{x}) \cdot \mathbf{E}(\mathbf{x}')}{\|\mathbf{E}(\mathbf{x})\| \, \|\mathbf{E}(\mathbf{x}')\|} \tag{10}$$

We further analyze the computed distances using the following two ways:

1. We compute the average of the three distances (by Equation 7) - $\mathrm{D}_1(q, i_1)$, $\mathrm{D}_1(q, i_2)$, $\mathrm{D}_1(i_1, i_2)$ - and plot the distribution of these mean values to show patterns.
2. We normalize the distances (by Equation 8) using the ratios $\mathrm{D}_2(q, i_1)/\mathrm{D}_2(i_1, i_2)$ and $\mathrm{D}_2(q, i_2)/\mathrm{D}_2(i_1, i_2)$, and plot these normalized values to reveal potential patterns.

The results from dense vector-based methods serve as baselines for comparison.

**Results.** Figure 5 presents the results using our path kernel-based method (with SAE), as well as two dense vector-based methods: one using SFR-Embedding-Mistral and the other using LLama-3.3-70B-Instruct. The horizontal axis shows the average distance assigned to each sample, computed as $\overline{(\mathrm{D}_1(q, i_1), \mathrm{D}_1(q, i_2), \mathrm{D}_1(i_1, i_2))}$, for both ambiguous questions and unambiguous questions in the AMBROSIA dataset. Moving along the x-axis from left to right corresponds to increasing average distance.

The vertical axis represents the absolute frequency, i.e., the raw number of observations falling into each of the 40 equal-width histogram bins. Superimposed on the histogram bars are kernel density curves, scaled so that their peaks align with the same frequency units. This allows for a direct visual comparison between the smooth density estimates and the discrete histogram counts.

As shown in the figure, our method results in fewer overlapping samples (27.5%) between ambiguous and unambiguous questions compared to the dense vector-based methods. Specifically, when using the x-coordinate of the intersection point of the red and blue density curves as a threshold for distinguishing ambiguous from unambiguous questions, the detection accuracies are as follows: Path kernel-based method (with SAE): 86.25%, Dense vector method with SFR-Embedding-Mistral: 70%, and Dense vector method with LLama-3.3-70B-Instruct: 77.75%. As a comparison, the Zero-shot accuracy of the Llama3-70B is 46.31%. We also visualise the distance relationship in Appendix B.2.

## 4.2 AGENTIC TOOL CALLING

**Experiment Settings.** We evaluate our tool-calling framework (Figure 4) on the Gorilla dataset (Patil et al., 2024). This multi-faceted benchmark contains about 1.6K ML-oriented API call templates sourced from HuggingFace, TorchHub, and TensorHub. The dataset includes training and test sets, as well as API collections that support retrieval-augmented generation (RAG). We analyzed the API call results (including both the API calls and their domains) and found that ambiguity is a major factor contributing to performance degradation. See Appendix C.1 for an illustrative example.

We evaluate the performance of our framework on the Gorilla dataset using several baselines: the Gorilla base model (7B), the Gorilla fine-tuned model (fine-tuned on the TensorFlow Hub API

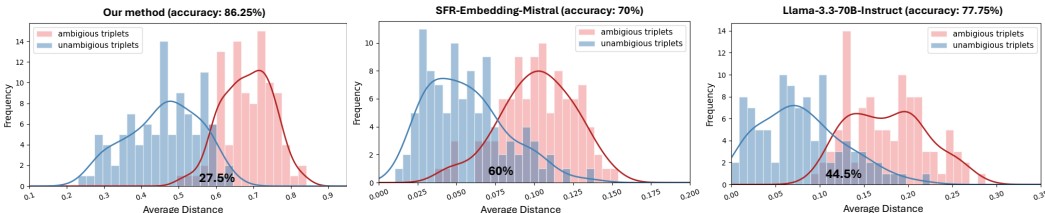

Figure 5: Distribution of average distances calculated using the path kernel method with SAE, and dense vector-based methods with SFR-Embedding-Mistral and LLama-3.3-70B-Instruct. A smaller overlapping area indicates a stronger ability to distinguish ambiguous from unambiguous questions.

dataset), and versions of fine-tuned Gorilla with BM25 and GPT-based retrievers. As the Gorilla base model is relatively small, for fair comparison, we also use a 7B model, Mistral-7B(The Mistral AI Team) with its sparse-autoencoder(Tyler Cosgrove) to implement our method. Additionally, we include SFR-Embedding-Mistral (Meng et al., 2024) as a baseline [2].

To predict the missing concepts in queries, we train a LightGBM model. We then evaluate the performance of our framework on the test data by using the predicted concepts to retrieve relevant API calls from the API collections. Considering the extra computational cost introduced by the sparse autoencoder (SAE), we do not retrieve all the concepts activated by the query. Instead, we select the top 50%, 30%, and 20% of the activated concepts, ranked by their activation values.

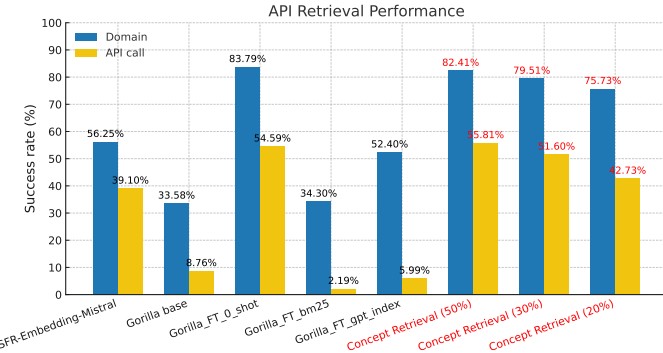

Figure 6: API bench Retrieval Results. Our methods are highlighted in red.

**Results.** Figure 6 shows the performance of our concept retrieval method compared to the baselines on the Gorilla TensorFlow Hub API bench. We evaluate the accuracy of identifying the correct API domains and retrieving the correct API calls. The red highlight shows the performance of our method, demonstrating that when using the top 50% of activated concepts, our approach achieves the highest accuracy in retrieving API calls. Accuracy of retrieving the correct domain is only slightly lower than Fine-tuned 0-shot Gorilla. We note that even when using only the top 20% of concepts (reduce the computational cost introduced by SAE), our method still outperforms all retrieval based baselines.

## 5 CONCLUSION

In this paper, we designed a novel concept-based method for ambiguity resolution in LLMs. Our method distilled concepts from ambiguous utterances and their associated interpretations, inferred the pattern of their difference in the latent space and leveraged the difference for ambiguity resolution. We demonstrated that out method outperformed baselines on the text-to-SQL task. We also gave a new method to improve LLMs' agentic tool calling performance through missing concept prediction. The method outperformed the SOTA in APIBench.

---

[2]SFR-Embedding-Mistral is ranked among the top 5 models on the MTEB leaderboard (Muennighoff et al., 2022).

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

## A    ALGORITHM: MEASURING SEMANTIC ENTROPY BY LLM

**Definition A.1** (Semantic Entropy). Given a prompt $x$, let $\mathcal{S}$ be the set of possible sequences (texts) and let $\mathcal{C}$ be a partition of $\mathcal{S}$ into semantic equivalence classes (meanings) $c \in \mathcal{C}$. The language model induces a distribution $p(s \mid x)$ over sequences $s \in \mathcal{S}$, which pushes forward to a distribution over meanings,

$$p(c \mid x) \;=\; \sum_{s \in c} p(s \mid x).$$

The *semantic entropy* of the model at $x$ is the Shannon entropy of this meaning distribution:

$$\mathrm{SE}(x) \;=\; -\sum_{c \in \mathcal{C}} p(c \mid x) \log p(c \mid x) \;=\; -\sum_{c \in \mathcal{C}} \left( \sum_{s \in c} p(s \mid x) \right) \log \left( \sum_{s \in c} p(s \mid x) \right). \tag{11}$$

At a high level the semantic entropy estimation involves three steps (Algorithm 1):

1. **Generation.** Given a prompt $x$, sample $N$ sequences $s_1, \ldots, s_N \sim p(\cdot \mid x)$ from the LLM.

2. **Clustering.** Embed each sequence $z_i = \mathcal{E}(s_i)$ and group the embeddings into semantic equivalence classes $C = \{C_1, \ldots, C_{|C|}\}$ via a clustering method $\mathcal{C}$ (cosine + agglomerative with threshold). Let $\ell_i$ denote the class label of $s_i$ and $C_k = \{i : \ell_i = k\}$.

3. **Entropy estimation.** Estimate the class masses $p(C_k \mid x)$ and compute entropy over meanings.

---

**Algorithm 1:** Measuring Semantic Entropy (Monte-Carlo class-average)

---

**Input:** Prompt $x$; num of samples $N$; generator $\mathcal{G}$; embedder $\mathcal{E}$; clustering procedure $\mathcal{C}$
**Output:** semantic entropy $H_{\mathrm{sem}}$, cluster probs $\{p_k\}$, labels $\{\ell_i\}$
```
// 1) Generate N samples and (optional) scores
```
**for** $i \leftarrow 1$ **to** $N$ **do**
$\quad \mid \quad y_i, s_i \leftarrow \mathcal{G}(x)$ ;            `// y_i:  text, s_i:  (avg) sequence log-prob`
```
// 2) Embed and cluster by meaning
```
$Z_i \leftarrow \mathcal{E}(y_i)$ for $i = 1..N$;
$\ell_1, \ldots, \ell_N \leftarrow \mathcal{C}(Z_1, \ldots, Z_N)$ ;            `// e.g., cosine + agglomerative`
```
// 3) Estimate cluster probabilities
```
Let $K$ be the number of distinct clusters among $\{\ell_i\}$;
**if** *using counts* **then**
$\quad \mid \quad n_k \leftarrow \sum_{i=1}^N \mathbf{1}[\ell_i = k]; \quad p_k \leftarrow \frac{n_k}{N}$ for $k = 1..K$;
**else**
$\quad \mid \quad$ `// probability-weighted`
$\quad \mid \quad$ `// Stabilize weights with log-sum-exp`
$\quad \mid \quad m \leftarrow \max_i s_i; \quad \tilde{w}_i \leftarrow \exp(s_i - m); \quad w_i \leftarrow \frac{\tilde{w}_i}{\sum_{j=1}^N \tilde{w}_j}$;
$\quad \mid \quad p_k \leftarrow \sum_{i=1}^N w_i \cdot \mathbf{1}[\ell_i = k]$ for $k = 1..K$;
```
// 4) Ensure numerical safety (clip and renormalize)
```
$\varepsilon \leftarrow 10^{-12}$;
$p_k \leftarrow \max(p_k, \varepsilon); \quad p_k \leftarrow \frac{p_k}{\sum_{j=1}^K p_j}$ for $k = 1..K$;
```
// 5) Calculate semantic entropy
```
$H_{\mathrm{sem}} \leftarrow -\sum_{k=1}^K p_k \log_b p_k \quad$ (default $b = 2$ for bits);
**return** $H_{\mathrm{sem}}, \{p_k\}, \{\ell_i\}$;

---

## B    AMBIGUITY DETECTION: PROMPT EXAMPLES AND ADDITIONAL RESULTS

### B.1 FEW-SHOT EXAMPLES USED IN PROMPTS TO INJECT CONCEPTS

Here we provide an example showing how concept-embodied examples can guide an LLM to generate diverse interpretations (Figure 7).

```
Example_1 = "question: What brands of agricultural machinery are available in each machinery
store\n
**interpretations**:
1. Which brands of machinery are equally available in all agricultural machinery stores?\n
2. For each agricultural machinery store, show which brands of machinery are available?\n"

Example_2 = "question: List the price of products sold in every duty-free shop.\n
**interpretations**:
1. For each duty-free shop, list the prices of all the products they sell.\n
2. What is the price of each product that is sold in all duty-free shops.\n"

instruction = f"Below question is ambiguous, and it has 2 interpretations. Please you generate
these 2 interpretations for this question. Here are two examples: Example 1: {Example_1}
Example 2: {Example_2} Now please generate 2 interpretations for below question. Don't answer
it is ambiguous or not, only answer the 2 interpretations. \n**Question**: " + question +
"\n**interpretations**:\n1. "
```

Figure 7: An example to prompt LLM to generate diverse interpretations.

### B.2 DISTANCE RELATIONSHIP

In Figure 8, we normalize the distances (Equation 8) for the distances between $q, i_1, i_2$ using the ratios $D_2(q, i_1)/D_2(i_1, i_2)$ and $D_2(q, i_2)/D_2(i_1, i_2)$, and plot these normalized values to reveal potential patterns. In this case, no concept mask is applied during distance calculation. Our goal is to examine distance patterns when using equation 8 with all activated concepts valid. For clarity, we visualize 100 samples for each case. We find that compared to unambiguous questions, interpretations for ambiguous questions are more concentrated and more symmetrically distributed in their distances to the questions.

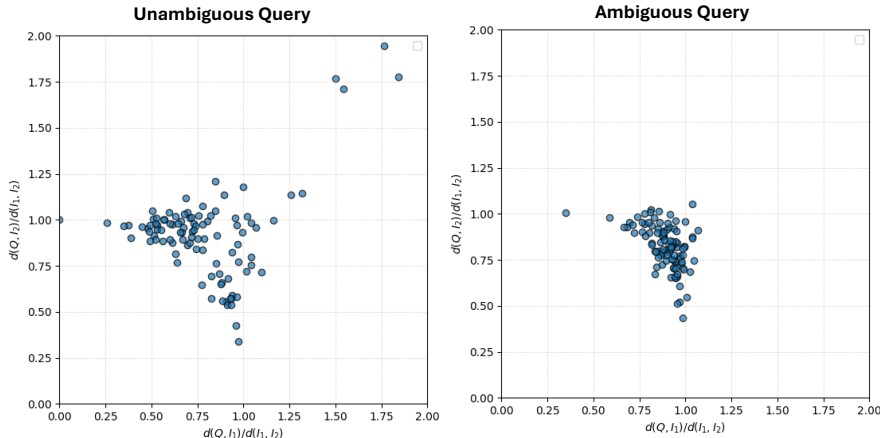

Figure 8: Normalized distances (Equation 8) for the distances between questions and their interpretations using the ratios $D_2(q, i_1)/D_2(i_1, i_2)$ and $D_2(q, i_2)/D_2(i_1, i_2)$, compared to unambiguous and ambiguous questions' distance triplets cluster to the center of the map.

Figure 9 shows the results of the baselines, we can see that distance calculations with dense vectors generated by both generation and embedding models cannot show the symmetric pattern of ambiguous questions and their interpretations. As such, we can not distinguish each data point is ambiguous or not by their measurements.

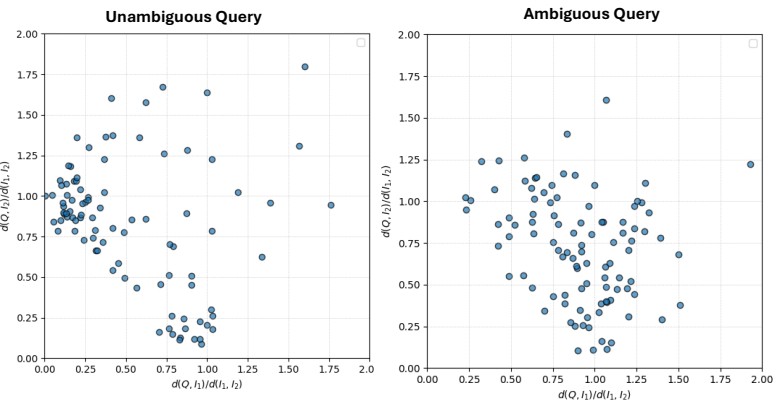

(a) Llama-3.3-70B-instruct model

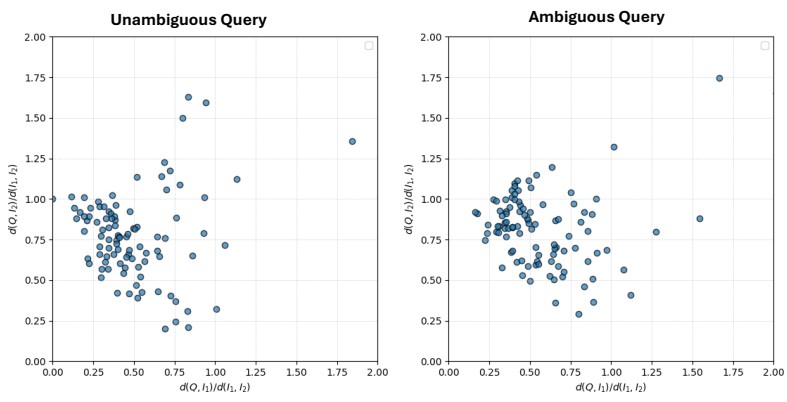

(b) SFR-Embedding-Mistral model

Figure 9: Normalized distances for the baselines.

### B.3 AMBIGUITY SENSITIVITY IMPROVEMENT OF LLMS WITH MISSING CONCEPT ADDITION

To demonstrate the connection between targeted concepts and ambiguity resolution, we conducted experiments with LLaMA3-70B on questions involving scope ambiguity. Below is an example (an ambiguous question and its interpretations):

Ambiguous question: "What brands of machinery are available in each machinery store?";
Interpretation 1: "Which brands of machinery are equally available in all machinery stores?";
Interpretation 2: "For each machinery store, show which brands of machinery are available."

As the missing concepts were found to correlate with tokens like "For," "each," "Which," "What," "all," "?", and "common", we first identified the concepts invoked by these tokens, and then manually increased the activation values of these concepts to 1.0. As results in table 1 show, we found that the accuracy of ambiguity detection on unambiguous questions increased from 39.8% to 60.5%. In contrast, when the same number of random concepts were activated instead, the accuracy dropped to just 0.2%. Although this will result in a 16.4% decrease in the ambiguous question detection accuracy, the overall accuracy will increase from 46.31% to 54.61%. This indicates that only targeted concept activation helps identify ambiguity, whereas randomly activating concepts only bring interference.

Table 1: Effect of concept activation on ambiguity detection and overall accuracy.

| Metric | Baseline | Targeted activation | Random activation | vs. baseline |
|---|---|---|---|---|
| Unambiguous accuracy | 39.8% | 60.5% | 0.2% | +20.7 pp (targeted) |
| Ambiguous accuracy | 59.3% | 42.8% | — | −16.4 pp (targeted) |
| Overall accuracy | 46.31% | 54.61% | — | +8.30 pp (targeted) |

*Notes.* "pp" = percentage points. Unambiguous questions and ambiguous questions in dataset are unbalanced.

## C  AGENTIC TOOL CALLING: PROMPT AND CONCEPT MATCHING EXAMPLES

### C.1  AMBIGUOUS PROMPT EXAMPLES

**Instruction:** "Find out what's in the image taken by a wildlife photographer, so we can determine the *main subject* of the picture.\n###Input: An image taken by a wildlife photographer."

**The API assistant searched:**
{"**domain**": *"Image object detection"*,
 "**framework**": "TensorFlow Hub",
 "**functionality**": "Detect objects in images",
 "**api_name**": "model_id",
 "**api_call**": "hub.load('https://tfhub.dev/tensorflow/ssd_mobilenet_v2/2')",
 "**api_arguments**": ["model_id"],
 "**python_environment_requirements**": ["tensorflow", "tensorflow_hub"],
 "**performance**": {"dataset": "COCO", "accuracy": "0.320"},
 "**description**": "A pre-trained TensorFlow Hub model for detecting objects in images using the Single Shot MultiBox Detector (SSD) architecture with MobileNet V2 as the base network."}

**Instruction:** "Find out what's in the image taken by a wildlife photographer, so we can determine the *object* of the picture.\n###Input: An image taken by a wildlife photographer."

**The API assistant searched:**
{"**domain**": *"Image feature vector"*,
 "**framework**": "TensorFlow Hub",
 "**functionality**": "Feature extraction",
 "**api_name**": "model_id",
 "**api_call**":
"hub.KerasLayer('https://tfhub.dev/google/imagenet/mobilenet_v2_100_224/feature_vector/4')",
 "**api_arguments**": {"model_id": "string", "input_shape": "tuple", "trainable": "boolean"},
 "**python_environment_requirements**": ["tensorflow", "tensorflow_hub"],
 "**performance**": {"dataset": "ImageNet", "accuracy": "Top-1 accuracy"},
 "**description**": "A pre-trained image feature vector model for image classification and transfer learning, based on MobileNetV2 architecture."}

### Ground truth

{"**domain**": *"Image classification"*,
 "**framework**": "TensorFlow Hub",
 "**functionality**": "Image classification using pre-trained model",
 "**api_name**": "imagenet_mobilenet_v2_100_224_classification",
 "**api_call**":
"hub.KerasLayer('https://tfhub.dev/google/imagenet/mobilenet_v2_100_224/classification/4')",
 "**api_arguments**": {"url":
"https://tfhub.dev/google/imagenet/mobilenet_v2_100_224/classification/4"},
 "**python_environment_requirements**": {"tensorflow": ">=2.0.0", "tensorflow_hub": ">=0.12.0",
"numpy": ">=1.19.5", "PIL": ">=8.3.2"},
 "**performance**": {"dataset": "ImageNet", "accuracy": "71.8%"},
 "**description**": "A pre-trained image classification model using MobileNetV2 architecture on ImageNet dataset with 100% depth and 224x224 input size."}

Figure 10: An example illustrating ambiguity in agentic tool calling (from the Gorilla dataset, using the Gorilla model as the API assistant). The red highlight marks differences in the instructions. Minor changes to the instruction can steer the LLM's answer, and may even shift the domain of the returned API.

### C.2  A CONCEPT MATCHING EXAMPLE

Figure 11 illustrates how the concepts that involve which activated by the input question and which predicted by pre-trained predictor are matched to those in the structured API document through the union joint operator.

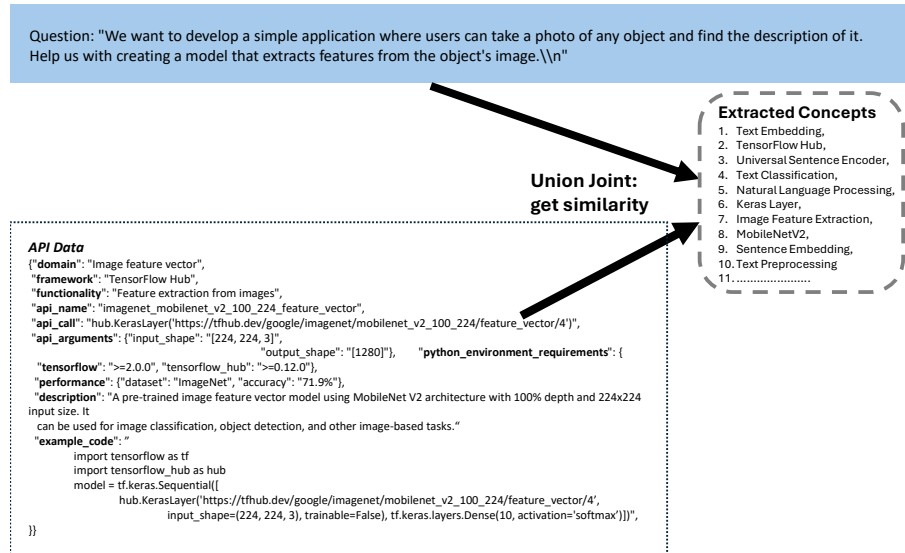

Figure 11: An example for getting similarity for extracted concepts by union joint.

## D  LIMITATIONS

Our method was evaluated on limited datasets. While results on both ambiguity and API datasets demonstrate its effectiveness, these datasets cover only a subset of known ambiguity scenarios, leaving it unclear whether our interpretation-generation method generalizes to other types of ambiguity in natural language. Investigating this question is left for future work.

## E  COMPUTING RESOURCES

Our experiments were conducted on four NVIDIA H100 GPU node, each with 96GB memory.

