# OpenReview forum: "Ambiguity in LLMs is a concept missing problem"
_ICLR.cc/2026/Conference — ICLR 2026 Conference Withdrawn Submission_

### Official Review · Reviewer_tKNc · 2025-10-30

**Soundness:** 3
**Presentation:** 2
**Contribution:** 3
**Rating:** 4
**Confidence:** 4

**Summary:**

This paper proposes leveraging a subset of SAE representations for ambiguity detection, instead of relying on conventional dense embeddings. The authors first demonstrate, through intuitive examples, that large language models (LLMs) struggle to generate diverse interpretations for ambiguous questions, and that slightly modifying the input (e.g., by introducing a [MASK] token) can help alleviate this limitation. Building on these empirical observations, the authors propose using SAE embeddings to measure the distance between different interpretations. Experimental results show that the proposed method outperforms standard dense embeddings on two tasks: text-to-SQL and tool calling.

**Strengths:**

- This work highlights the lack of research on the representational differences of ambiguous text and tackles this underexplored yet important problem, which I also find crucial.
- The case studies presented in Section 3.1 are intuitive and help readers clearly understand the problem that this work aims to address.
- The idea of using representations derived from SAEs for ambiguity detection is relatively novel and valuable, as it connects theoretical probing studies using SAEs with practical scenarios where models must identify the ambiguity of a given question.
- When evaluated on two datasets, AMBROSIA and Gorilla, the proposed method demonstrates reasonable performance compared to typical embedding-based approaches.

**Weaknesses:**

- First of all, I would like to mention that I basically like the overall idea of this work. However, given the current state of the draft, there are several aspects that need improvement to better clarify the paper’s contributions.
- There is room for clarification and refinement in the notations and equations. Some notations appear to be directly adopted from previous works without sufficient caution. For example, in Eq. (2), what does $\mathbf{w}$ represent? I could not find a definition in the paper. Similarly, SAE in Eq 2 $\rightarrow$ $f_\text{SAE}$ ?
- It is also unclear how the mask $M$ is constructed—what proportion of masking is applied? The explanation seems rather abstract, remaining at a conceptual level without sufficient implementation detail.
- Regarding the Path State Approximation, using a linear approximation might be reasonable and perhaps the only feasible option. However, there is no discussion of the potential risks of this abstraction. Most researchers would agree that the training path is typically non-linear, so additional discussion on the implications of approximating such a non-linear trajectory with a linear one would be valuable.
- Moreover, the experimental section feels somewhat limited. The results are presented for only two tasks, without detailed discussion or ablation studies on the components of the proposed method. Adding such analyses could significantly strengthen the paper.
- Studies on ambiguity handling generally focus on (open-ended) question answering benchmarks such as NQ and TriviaQA. I appreciate that this work takes a different direction by applying the proposed method to more diverse and underrepresented tasks, such as text-to-SQL and tool calling. However, since most prior methods are evaluated on QA datasets, it would make the comparison more convincing if the proposed method’s performance were also reported in that setting.
- It is also unclear whether simply averaging all distances—$D(q, i_1)$, $D(q, i_2)$, and $D(i_1, i_2)$—is sufficient. Intuitively, the effect of each term may be diluted by the others. Could there be a more principled way to handle this part? Perhaps additional case studies could help justify that the current simple averaging approach is indeed reasonable (or reveal that it may not be).
- It would also be valuable if the draft explicitly discussed the pros and cons of the proposed approach. In practice, training SAEs from scratch is quite cost-intensive, and most studies instead rely on utilizing pre-trained and open-sourced SAEs available for only a limited number of models. Furthermore, incorporating SAEs inevitably introduces additional computational costs, as briefly mentioned in Section 4.2. Clearly addressing this efficiency aspect would make the paper more transparent and substantively stronger.
- It appears inconsistent to rely on vanilla LLMs for generating interpretations in Section 4.1, considering that Section 3.1 already shows their limited capability in handling the task.

**Questions:**

Please refer to the Weaknesses section.

---

### Official Review · Reviewer_BofT · 2025-11-01

**Soundness:** 2
**Presentation:** 2
**Contribution:** 2
**Rating:** 2
**Confidence:** 3

**Summary:**

This paper investigates the origin of ambiguity in LLM and proposes that it arises from missing concepts in the model’s latent space. Building on recent work in mechanistic interpretability, the authors employ a Sparse Autoencoder (SAE) to extract human-interpretable latent concepts from LLM activations. They then introduce a path-kernel formulation that measures distances between questions and their possible interpretations by integrating gradients along the parameter trajectory of the model. Based on this hypothesis, the paper develops a concept predictor that identifies and injects missing concepts to mitigate ambiguous behavior. Experiments are conducted on the AMBROSIA dataset for ambiguity detection and on Gorilla TensorFlow Hub for tool-calling under ambiguous instructions. The proposed SAE-based path kernel shows higher accuracy than dense embedding baselines and demonstrates improved API retrieval performance when missing concepts are predicted and supplied. Overall, the work presents an interesting conceptual connection between ambiguity, interpretability, and kernel methods, and serves as a proof-of-concept study toward understanding and reducing ambiguity in LLM reasoning.

**Strengths:**

- The paper presents an interesting and novel view that ambiguity in LLMs can be interpreted as a missing concept problem, linking ambiguity detection with model interpretability.
- Integrating sparse autoencoders with path kernels is a creative idea that provides a new way to measure semantic differences beyond dense embeddings.
- The proposed method achieves clear performance gains on AMBROSIA and Gorilla benchmarks, showing empirical value beyond conceptual novelty.

**Weaknesses:**

1. Incomplete methodological exposition.
Section 3.4 (“Predicting Missing Concepts to Mitigate Ambiguity”) is poorly explained. The paper does not specify how labeled data are obtained, what features are used as input to the concept predictor. The integration between the predictor and retrieval module (“union joint”) is vague, and the role of the path kernel in this stage is unclear. Figure 4 is also oversimplified, leaving the data flow between modules undefined and the overall pipeline difficult to reproduce.

2. Speculative core hypothesis without rigorous validation
The central claim that “ambiguity can be treated as a missing concept problem” (Section 3.1) is primarily a theoretical conjecture rather than an empirically verified mechanism. The paper demonstrates this connection through a single case study with the [MASK] token (Figure 1 and Section 3.1), but does not include systematic experiments or statistical evidence showing that missing-concept activation consistently correlates with ambiguity across datasets. The insight is interesting but remains under-supported.

3. Unclear definition and computation of semantic entropy
Section 3.2 introduces “semantic entropy” to quantify ambiguity but does not clearly explain how it is computed in practice.

4. Simplistic approximation of training trajectory
The approximation of the training path in Section 3.3 by a “straight-line interpolation in parameter space” (Eq. 7–8) is a strong simplifying assumption. Since the entire path kernel formulation hinges on gradient-trajectory alignment, this approximation could drastically alter kernel behavior. The paper should have included experimental validation comparing results with true gradient trajectories (even on smaller models) to justify the substitution.

5. Unclear parameter initialization and reproducibility gaps
The paper references initialization parameters θ^((0))  in the path kernel formulation but does not specify how these parameters are obtained, whether they come from pretrained SAE checkpoints or random seeds, or whether results are stable across different initializations. Such omissions raise reproducibility concerns, especially because kernel values depend on initialization when straight-line interpolation is used.

6. Limited experimental diversity and external validation
The experiments are primarily conducted on AMBROSIA and Gorilla TensorFlow Hub datasets. Both are specialized and relatively small in scope. There is no evidence that the proposed method generalizes to other ambiguity types (e.g., lexical, referential, pragmatic) or to other tasks (beyond SQL parsing and tool calling). A cross-domain evaluation or at least a synthetic control experiment would significantly strengthen the claims.

7. Missing ablation and sensitivity analyses
Important hyperparameters, such as the number of interpolation steps, concept-mask size, or LightGBM predictor depth, are not explored. The paper also lacks ablations to disentangle the effects of (a) path kernel vs. dense embeddings and (b) concept masking vs. full concept sets. Such analyses are necessary to verify that observed gains stem from the proposed mechanism rather than incidental tuning.

8. Presentation and clarity issues
The overall exposition suffers from poor organization and missing transitions between conceptual sections. Figures 2–4 are not fully explained in the text, and some equations are introduced without definitions of symbols or variable ranges. The paper would benefit from a clearer flow of ideas, with intuitive examples accompanying mathematical sections.

**Questions:**

1.	On dataset diversity and coverage
Your experiments focus mainly on AMBROSIA and Gorilla datasets, which represent scope and structural ambiguity. How would your method handle other ambiguity types, such as referential or pragmatic ambiguity? Do you expect the same “missing concept” mechanism to hold there?
2.	On reproducibility details
Could you provide essential implementation details, such as SAE layer configuration, number of interpolation steps in the path kernel, LightGBM feature inputs, and training data sources for the concept predictor, so that others can reproduce your results?
3.	On hypothesis validation
The paper posits that “ambiguity = missing concept” in the latent space, but this is demonstrated with only one case and activation examples. Can you provide quantitative evidence (e.g., correlation between missing-concept activations and ambiguity labels) to substantiate this causal link?
4.	On the path kernel approximation
Since you replace the true gradient path with linear interpolation, have you evaluated how this approximation affects results? Would performance degrade or patterns change if a real or simulated training trajectory were used?
5.	On the semantic entropy metric
Could you clarify how semantic entropy is computed in practice, which embeddings, clustering thresholds, and sample sizes are used, and whether you observed sensitivity to these parameters?
6.	On ablation and robustness
Have you performed ablations to isolate the contribution of (a) concept masking vs. full concept sets and (b) path kernel vs. standard cosine similarity? Such analysis could confirm that observed gains indeed arise from your proposed mechanisms.
7.	On the stage of validation
You describe the approach as a “proof-of-concept.” What would be the next step toward turning it into a more generalizable framework, e.g., scaling to larger or more diverse corpora, or testing in real interactive ambiguity-resolution settings?

---

### Official Review · Reviewer_MqEV · 2025-11-01

**Soundness:** 1
**Presentation:** 1
**Contribution:** 2
**Rating:** 2
**Confidence:** 3

**Summary:**

This paper presents a method from distinguishing ambiguous from unambiguous queries. The method involves (2) prompting an LLM to generate two interpretations $i_1$ and $i_2$ for a given query $q$, then (2) computing the distances $D(q,i_1),D(q,i_2), D(i_1,i_2)$ using a "path kernel-based method (with SAE)". The average of these three distances is essentially higher for ambiguous queries, and lower for unambiguous queries. In experiments, they show that their detection method achieves 86% accuracy on the AMBROSIA dataset, outperforming simpler distance metrics based on LLM-produced dense embeddings. Moreover, they show that by using the generated interpretations to support agentic tool calling, the overall performance can be improved.

**Strengths:**

1. This paper proposes a method for ambiguity detection, and show that there are benefits both in detecting ambiguity and augmenting agentic workflows with alternative interpretations of the user query.

**Weaknesses:**

1. Overall, the clarity of the technical writing is weak, making it difficult to understand what was actually done.

- The following two interpretations have the same flaw to me, which is that they are both ambiguous themselves. How come the first one is considered wrong and the second one is correct?

"Show all gate agents and pilots who speak Spanish"
and
"Show all gate agents and pilots who are Spanish-speaking"

- L. 176: How was the SAE trained? How is it decoding concepts in natural language?
- How did you clamp activation value of a given concept, and why does the figure instead say that it's *activating* the concept? In particular, the activation values where, and how do you know they correspond to some concept?
- §3.2: How do you calculate semantic entropy? Where are the 20 queries from?

2. Most of the paper (e.g., Figure 1, the motivation example in §3.1) seemed to say that a query is ambiguous when the distance between the two interpretations is larger than the distance between the query and either interpretation. However, if I understand correctly, the experiments actually use the straight average of the three distances as an indicator of ambiguity.

3. It is really difficult to understand whether the significant added complexity to the method was necessary or made sense as a solution to the problem of ambiguity detection. Perhaps some higher-level intuition for how the method works would be beneficial.

**Questions:**

NA

---

### Official Review · Reviewer_iMih · 2025-11-02

**Soundness:** 2
**Presentation:** 2
**Contribution:** 3
**Rating:** 4
**Confidence:** 4

**Summary:**

This paper presents an approach for characterizing representational differences of ambiguous text within latent space and identifies ambiguity before mapping the text to structured data. To address sentence-level ambiguity, the authors examine the relationship between ambiguous questions and their corresponding interpretations. It also introduces an alternative metric based on a path kernel over conceptual structures and proposes a method to enhance the performance of large language models (LLMs) in handling ambiguous agentic tool-calling tasks by predicting missing concepts.

**Strengths:**

1. This paper posits that linguistic ambiguity stems from missing conceptual representations within the latent space of large language models (LLMs) and introduces a distance metric to improve interpretability while capturing specific semantic patterns.
2. This paper identifies systematic patterns that effectively differentiate ambiguous questions from unambiguous ones.
3. This paper further proposes a comprehensive framework aimed at enhancing LLM performance in managing ambiguous agentic tool-calling tasks through the prediction of missing concepts.

**Weaknesses:**

1. The evaluations are conducted solely on the AMBROS (text-to-query) and Gorilla (tool-calling) datasets, which raises two primary concerns: (1) this limited scope renders the study somewhat fragile and lacking in coherence, and (2) the absence of broader testing on additional QA tasks restricts the generalizability of the approach.
2. The evaluation is further weakened by the omission of key baseline comparisons. Specifically, methods referenced and critiqued in the Introduction section, such as ReACT, ClarifyGPT, and the approaches by Kamath et al. (2024) and Saparina & Lapata (2025), are not included in the experimental analysis. Without these comparative baselines, it is difficult to determine whether the proposed justifications and methods provide a genuine performance advantage.
3. The paper contains presentation issues, such as misuse of `\citep{}` and `\citet{}`, and a lack of space before citations.

**Questions:**

See "Weaknesses."

---

### Note · Authors · 2025-12-03

I have read and agree with the venue's withdrawal policy on behalf of myself and my co-authors.